# Perception and Understanding of Guideline Daily Amount and Warning Labeling among Mexican Adults during the Law Modification Period

**DOI:** 10.3390/nu14163403

**Published:** 2022-08-18

**Authors:** Ismael Campos-Nonato, Manuel A. Cervantes-Armenta, Selene Pacheco-Miranda, Amado D. Quezada-Sánchez, Alejandra Contreras-Manzano, Simón Barquera, Jorge Vargas-Meza

**Affiliations:** 1Center for Nutrition and Health Research, National Institute of Public Health, Av. Universidad 655 Col. Santa María Ahuacatitlán, Cuernavaca C.P. 62100, Morelos, Mexico; 2Center for Research in Evaluation and Surveys, National Institute of Public Health, Av. Universidad 655 Col. Santa María Ahuacatitlán, Cuernavaca C.P. 62100, Morelos, Mexico

**Keywords:** warning label, front-of-pack labeling, politic, implementation, Mexico

## Abstract

Front-of-pack labeling (FOPL) is a tool that enables consumers to compare foods and select healthier options. Due to low understanding of the Guideline Daily Amount (GDA) labeling among Mexicans, a law was implemented in October 2020 that modified the FOPL to a warning labeling (WL) system. The purpose of this study was to compare the perception and understanding of GDA and WL during the law modification period. We conducted a panel design with two measurements: (1) using GDA label (September 2020) and (2) using WL (October–November). We estimated differences in GDA vs. WL through multinomial logistic regression models and changes were measured through predictive margin contrasts and Wald tests. When comparing the same products with different labels, the participants reported that it would be unlikely/very unlikely that they would consume products packaged with the WL (81.5%; 95%CI: 79.2, 83.8) compared to those with GDA (24.2%; 95%CI: 21.7, 26.7). Consumers’ perception was that the quantities of packaged products they should consume was small or very small when they used the WL (93.8%; 95%CI: 92.4, 95.5) compared to GDA (41.6%; 95%CI: 39.7, 44.6). When comparing food groups, participants were more confident about choosing healthy products when using the WL compared to the GDA. During the implementation of WL in Mexico, the studied population had a better perception and understanding of less healthy packaged foods when using WL, compared to the GDA label.

## 1. Introduction

Processed and ultra-processed products are characterized by high energy density, sugar, sodium, saturated fat, trans fat, and a low content of nutrients that are beneficial to health [1,2]. Exposure to ultra-processed foods is associated with at least one adverse health outcome such as obesity, cardio-metabolic risks, cancer, type 2 diabetes, cardiovascular disease, depression, and all-cause mortality [3].

Informing consumers about the nutritional content of food products through simple and easy-to-understand front-of-package labeling (FOPL) is an intuitive and useful tool to avoid selecting unhealthy options that negatively affect health [4]. Warning the population about the excessive amounts of critical nutrients (i.e., added sugars, saturated fat, sodium, etc.) in processed and ultra-processed foods based on a determined nutrient profile criteria [5,6] has implications for public policies that include the need to establish a FOPL, regulate its marketing, and establish regulations for unhealthy products [7].

In addition, FOPL can encourage the selection of healthier food options, promote the substitution of products with low nutritional quality, and help increase the consumption of fresh and natural food options [8]. The format and design of the FOPL influences the consumer’s ability to distinguish the least healthy products, and also has the potential to affect their purchasing decisions [9,10]. However, the effect of labels is mediated by consumer acceptability and understanding of the label [11], as well as nutritional knowledge and interest in the nutritional quality of packaged foods [10].

Different types of FOPLs exist around the world based on local evidence on acceptability, comprehensibility and effect on decision making at the point-of-sale [12]. For example, those that highlight whether the food product is healthy according to certain criteria or nutrient profiles such as Health Star Rating, Keyhole and Options; those that warn whether food products have excessive nutrients of interest such as Warning Label (WL) and Multiple Traffic Light; and those that indicate the contribution of nutrients to the diet such as the Guideline Daily Amount (GDA) [12]. 

During 2014, the Mexican government implemented GDAs to promote healthy dietary choices and inform consumers [13]. However, the selection of this labeling was not evidence-based and Mexican consumers have shown low understanding and acceptability towards this label [14]. For that reason, in October 2020, Mexico implemented a law (NOM-051) that established that pre-packaged beverages and foods that exceed a threshold must have a WL of “Excess” calories, sugars, saturated fat, trans fats, or sodium according to the addition/content of each packaged food [15]. In addition, it included an innovative format of numeric WL for products with packages smaller than 40 cm2 or returnable packages. Compared to the other countries that have implemented this system, Mexico included two precautionary disclaimers for children regarding non caloric sweeteners and added caffeine [15]. 

The perception and understanding of WL in the Mexican population was evaluated before the implementation of NOM-051 [11,16], but there is no evidence pertaining to the period after the regulation became mandatory in the country. Therefore, the objective of this study is to evaluate the perception and subjective understanding among Mexican consumers of the GDA label and the WL implemented in Mexico. 

## 2. Materials and Methods

We specified a panel design with two measurements: before and after the mandatory implementation of NOM-051. Data was obtained through an online questionnaire. The first evaluation assessed the GDA label from September 1 to 15, 2020, one month before NOM-051 went into effect. The second evaluation assessed the WL 15 days after NOM-051 went into effect (15 October 2020, until November 2020). The effects proposed by the conceptual model of Taillie et al. were considered, which states that for a WL to be used to acquire and consume a healthy product, the WL has to have an impact on behavioral intentions, perceptions of health risk, and must be understandable [17]. 

### 2.1. Recruitment and Procedures

We trained undergraduate students from ten different universities across the country to carry out the study procedures. Two members of our research team (JVM and MACA) trained research assistants on how to recruit prospective participants. To identify participants, the research assistants explained the study objectives, the stages of the study, and invited them to be part of the study. Each research assistant was instructed to recruit a minimum of 20 participants in any of their primary networks of people close to them and to follow up on their participation after being included in the study. It was suggested that people be recruited according to their employment and living areas to identify low- and middle-income groups in Mexico. Previous training was provided on how to identify areas of different socio-economic levels (SES).

**First Evaluation (1E).** The questionnaire was self-administered. To answer 1E questions, the researchers accessed a unique web address where our tool was hosted on the RedCAP platform using any mobile device (smartphone, laptop or desktop tablet with Internet access). At the beginning of the recruitment, the eligibility of potential participants was evaluated using a 4-question screening questionnaire. Inclusion criteria were (1) adults (>18 years); (2) who shop (at supermarkets or convenience stores) at least once every fortnight for foods and beverages such as sugary drinks (i.e., soda), cereals (i.e., boxed biscuits or cereal), salty snacks (i.e., peanuts or chips); (3) no conflict of interest (that they or any of his/her immediate family members do not work in the health area or in any industrialized food and beverage company); and (4) that they could read and write. If a participant did not meet the selection criteria, the platform classified the participant as ineligible. If the participant was eligible, written informed consent and a written signature were requested on the web platform. Subsequently, sociodemographic information was obtained and knowledge in health and nutrition was evaluated. Regarding labeling, a questionnaire about perception and an exercise to test the subjective understanding of the GDA label (Figure 1a), as well as five products with the same label on the front of the packaging were used.

**Second Evaluation (2E).** After 15 days of the implementation of NOM-051, E1 participants were identified to answer an online questionnaire for E2. Each research assistant contacted their participants through the list they developed in the first phase. In this evaluation, the questions about subjective perception and understanding were used with the WL octagons (Figure 1b). In addition, questions about the warning disclaimer for sweeteners and caffeine (Figure 1c,d) were included to assess perception, knowledge, and risks of consumption.

**Materials.** Five products of low nutritional quality from different food categories (Sweet drink with sugar, salty snacks, cereals, dairy products, and ready-to-eat foods) were randomly selected from a database compiled by the National Institute of Public Health of Mexico between 2015 and 2016 [18]. The selected products were turned into dummies to avoid aspects of familiarity with the product and brand; however, the dummy labels resembled the graphic design of existing brands. The GDA label was affixed as shown on the original product. The WL were assigned to each product according to the nutritional information on the original packaging, applying the first phase of the nutrient profile of the NOM-051 [15]. The Appendix A shows the different products used and the nutritional information according to the labeling conditions.

### 2.2. Perception of the FOPL

For each evaluation, the perception was tested showing the image of the label, as well as in five products with the label corresponding to the evaluation. We used a questionnaire obtained from an international study which explored consumers’ perceptions of five FOPL labels in 12 countries [19]. To assess perceptions about the FOPL scheme, participants were asked to answer 11 adapted questions. To test perception, participants were asked to rate the label using adapted six questions. Different Likert scales were used (i.e., 1 to 3, or 1 to 7) (see Appendix A).

### 2.3. Subjective Understanding of FOPL

We tested subjective understanding in each evaluation (using GDA and WL) with a series of five exercises per participant, corresponding to the five products. Observing the critical nutrients (sugar, trans fat, saturated fat, and sodium) and energy that both FOPL showed, the participants had to select those with high or excessive amounts. For each product, two images were shown: one showed the complete product, and the second image showed the amplified label of the product. Figure 2 shows how the products were displayed for each of the evaluations. In both evaluations (1E-2E), the exercises followed the same dynamic (see Appendix A).

### 2.4. Knowledge and Perception of Non-Caloric Sweeteners and Caffeine Disclaimers

Knowledge about non-caloric sweeteners was evaluated, as well as perceptions towards caffeine and the non-caloric sweetener disclaimer implemented in NOM-051 using fifteen adapted questions with different Likert scales [20] (see Appendix A).

### 2.5. Covariates

Using a validated questionnaire, we collected sociodemographic variables such as sex (male, female), age (continuous), self-declared weight and height (Kg and mts), education (none, primary, secondary, high school, university, postgraduate), nutritional knowledge, and interest in own health (both Likert scale: 0 = nothing to 3 = a lot), as well as health variables such as previous diagnosis of hypertension, diabetes, overweight, obesity, high cholesterol or high triglycerides if they answered “yes” to the question “Has a doctor ever told you that you have...”. SES was estimated according to the Mexican Association of Market Intelligence (AMAI, acronym in Spanish) [21]. This questionnaire estimates the level of SES based on the score of the answers to six sociodemographic questions.

### 2.6. Determination of Sample Size and Sampling Procedure

The sample size was determined to obtain specific levels of precision for estimates of mean intra-subject changes of numeric standardized scores and assuming a non-response rate of 10%. We planned a total sample size of 1200 which correspond to 1080 observations after eliminating non-response. The attained margins of errors (standardized scale) were 0.07, 0.06 and 0.05 for pre-post correlations of 0.3, 0.5 and 0.7, respectively. 

### 2.7. Ethical Considerations

All participants received and signed an informed consent form approved by the Institutional Review Board (IRB) of the National Institute of Public Health in Mexico (INSP). The original protocols have the approvals of the Ethical and Research Commissions of the INSP, with Commission Number 1122 and 1401. The study was in accordance with national guidelines (Declaration of Helsinki).

### 2.8. Data Analysis

A descriptive statistical analysis was conducted for the general characteristics of the participants according to age in categories as defined by tertiles. Categorical variables were described as frequencies and percentages. 

To estimate changes in perception of FOPL, each statement scale was recoded into three categories. To estimate the perception of FOPL and subjective understanding using different products, the scales were recoded into two categories for each question.

We estimated differences in perception of FOPL through multinomial logistic regression models for three-category outcomes and logistic regression models for binary outcomes. In all models, the linear predictor included age group indicator variables and indicator variables from the 2E study stage and all their interactions. Additionally, we adjusted for SES, knowledge of nutrition, and BMI in all models. Covariate-adjusted proportions were obtained in total and by age group and predictive margins were calculated [22]. In addition, changes were obtained through predictive margins contrasts and Wald tests were applied.

To describe the knowledge and perception of non-caloric sweeteners, as well as the understanding and use of labels for non-caloric sweeteners and caffeine, proportions, 95% confidence intervals, and chi-square tests were used. All the analyses were performed using Stata version 15 (College Station, TX, USA). We set the significance level for all tests at 0.05.

## 3. Results

During 1E, a total of 1640 questionnaires were obtained. However, a total of 1083 participants answered the questionnaire in 2E (non-response rate 34%). The mean age of the participants was 36.5 years, 60% were females, 69.7% belonged to a low socioeconomic level, 51.3% had a bachelor’s degree or above, and 57.5% had overweight and obesity (Table 1).

Table 2 describes the adjusted percentages and the differences in percentage points of the perception between GDA and WL labels. When comparing the same products with different labeling, the participants reported that it would be unlikely-very unlikely for them to consume products packaged with the WL (81.5%; 95%CI 79.2, 83.8) compared to the GDA (24.2%; 95%CI 21.7, 26.7). Consumers’ perception was that the quantities of packaged products they should consume was small or very small when they used the WL (93.8%; 95%CI 92.4, 95.5) compared to GDA (41.6; 95%CI 39.7, 44.6). On the other hand, the WL octagons were easier to identify (difference of 42.8 percentage points) and to understand (difference of 39.5 percentage points) compared to the GDA (*p* <0.001).

When comparing food groups, participants were more confident about choosing healthy products when using the WL compared to the GDA (Table 3). Participants perceived the products with WL as less attractive and less healthy. They also reported a lower probability of purchase, a lower frequency of consumption, and perceived WL to be more informative, in comparison to how they perceived the same products when the GDA was displayed on the package. Across food groups, the greatest differences between the GDA and WL were found for ready-to-eat cereals, sugary drinks, and salty snacks (*p* < 0.001). Additionally, WL allowed for better identification of critical nutrients than the GDA, showing the greatest differences for ready-to-eat cereals (74.9 + 1.41), salty snacks (54.8 + 1.68) and dairy (39.1 + 1.71) (*p* < 0.001) (see Appendix A).

The knowledge of non-caloric sweeteners, as well as the perception of the disclaimers for non-caloric sweeteners and caffeine was evaluated. Participants reported that they strongly agreed or agreed that non-caloric sweeteners are present in many soft drinks and juices (52%), that children should not consume foods and beverages with non-caloric sweeteners (45%), and that the non-caloric sweetener disclaimer would help them decide whether to buy a product (52%). In addition, 74% reported that they would almost never give a child a product with the non-caloric sweetener disclaimer, while 69% reported that a product with this disclaimer should be consumed in very small amounts. On the other hand, for the caffeine disclaimer, 82% reported that they would almost never give a child a product with this disclaimer, while 65% reported that a product with this warning disclaimer should be consumed in very small amounts (see Appendix A).

## 4. Discussion

We found that participants better perceive and understand the WL compared to the previously implemented FOPL (GDA) when only the images of the labels are displayed. When the labels were used on the front of packaged foods, the differences of all the evaluated points of perception and comprehension in WL were greater compared to GDA, mainly in the categories of ready-to-eat cereals, sugar-sweetened beverages, and ready-to-eat foods. Furthermore, the results of this study contribute to the evidence about the understanding of WL, reinforcing that WL are effective at helping consumers identify less healthy products and understand their associated health risks.

Several studies have demonstrated similar results comparing these labels, showing that WL is a better choice for identifying foods with high critical nutrients [23,24,25]. Previous studies conducted in Mexico have also shown that WL is more objectively understandable compared to GDA, as they considered study participants were more likely to identify a less healthy product when using WL, compared to GDA [11,26]. Furthermore, the WL has been shown to require less time to identify these less healthy products compared to the GDA (11.9s vs. 15.3s) [11]. These results have also been observed among white, Latino and Mexican consumers. This study showed that participants were more likely to understand the WL compared to the nutrition facts table and GDA. It is important to highlight that some of these studies were carried out with a virtual store simulation [11,26], as well as an international survey [16]. Although our results do not show understanding in a real environment, it is possible that the population has already been exposed to the WL, since the information was collected after the NOM-051 had already been implemented. Therefore, our study reinforces that the understanding of the WL is greater than for the GDA. 

Various documents have reported that the implementation of the GDA label in 2014 was supported by the food industry without the participation of other groups such as academics and/or scientists. In addition to this, there is currently robust evidence that the GDA label is not a format that helps the Mexican population to select healthy foods [14,27,28,29,30], including those with chronic non-communicable diseases. [31].

Our results showed that perceived health risk associated with the product was higher when participants saw the WL compared to when they saw the product with the GDA label. A qualitative study with Chileans showed that after the WL was implemented (2016), mothers reported that the regulation changed perceptions of knowledge, attitudes and practices towards the consumption of healthy foods [24]. In Peru, the participants agreed that labeling could influence their purchase decision (48%), and most were in favor of the implementation of a WL (76%) and with the model or label design (63%) [32]. In Uruguay, an observational study indicated that immediately after WL implementation, the labels could reduce the selection of unhealthy foods since they caused a negative perception for the health of consumers [33].

Due to the above, several countries have considered the implementation of a WL system. Such is the case in Brazil that, during 2020, the National Sanitary Surveillance Agency (ANVISA) approved a warning system [34]. A recent study developed with a Brazilian population compared the perception of healthiness of various FOPL systems using nine products. The results indicate that the presence of the WL was the only one that significantly reduced the perception of healthiness for the nine products, compared to the control group (Traffic light label) [35]. The data in the articles cited above support the results of our study since the same participants evaluated the same products with different labels in two different moments, resulting in reduced perception of food healthiness <when WL were displayed. The perception of healthiness of a food product is influenced by numerous factors such as the information communicated. In general, it could be due to the shape and color of the packaging, the ingredients, food group category, organic origin, as well as the taste and other sensory characteristics of the product [36]. In this study, we tried to reduce other factors such as the familiarity of the product and brand, as well as other types of information on the packaging such as claims or characters so that the perception was direct towards the information shown in the FOPL. Therefore, our differences found in the healthiness of the products are mainly attributable to the label that was displayed on the package.

Regarding knowledge about non-caloric sweeteners, one third of the participants perceived non-caloric sweeteners as harmful to health and half of the participants knew that these additives are present in sugary drinks. This low acceptance has been described in previous studies. Non-caloric sweeteners are commonly used to restrict caloric intake from sugars, especially those who have a chronic condition such as overweight and diabetes [37]. Studies conducted with young adults in Canada and the United Kingdom show that the population perceives high fructose corn syrup and aspartame to be less healthy than table sugar [38], and they perceive a high risk among sweeteners [37]. This last finding was related to the reduction in the consumption of non-caloric sweeteners, which may be related to the “nature of the sugar” [37,38]. In accordance with the above, a subsample of this study who participated in a qualitative study reported that they did not understand what a non-caloric sweetener was, but when they identified a disclaimer on the products, they tried to avoid consumption and to avoid giving food with this disclaimer to minors. [39].

In the recent NOM-051, two precautionary disclaimers were added with the text: “CONTIENE EDULCORANTES, NO RECOMENDABLE EN NIÑOS” (Contains sweeteners, not recommended in children), as well as ““CONTIENE CAFEÍNA EVITAR EN NIÑOS” (Contains caffeine avoid in children). According to our results, these disclaimers can lead to a high perception of risk, as they reported not giving foods with disclaimers to children and consuming these products less frequently. This perception could also be related to prior knowledge of non-caloric sweeteners, as well as to the trust of the institutes that regulate the law [37]. For this reason, it is important to generate communication strategies to increase awareness among the population regarding the health risks of these two additives in children’s diets [40].

This study did not evaluate the potential effect that different aspects of the label would have had in the real world. However, it is known that WL are easy to understand and useful to modify the consumers´ misperception of products. In addition, it has been shown that WLs are useful even when accompanied by health/nutrition claims [41]; they can improve healthier purchasing decisions [26]; that they can reduce the sales of unhealthy products [42]; and they promote reformulation of products with low nutritional quality [43,44].

This study has some limitations. Our study does not include a representative sample of the Mexican population. However, this study considered a population from five different states, which included the different regions of the country and mostly individuals with low and middle SES. In addition, it is important to highlight that the population reported a high level of education (51% high school or higher), medium-high level of knowledge in nutrition (40%), and a high interest in their own health. These aspects are directly related to a greater understanding of the labels, which influence their perception of them [17]. However, the profile of the participants is similar to the SES of households in this country [45]. 

Another limitation is that the time between one assessment and another was approximately one month. During this time, participants could have been informed about nutrition labels or exposed to different sources of information about the implementation and use of the WL. However, during the implementation of the labeling law, there was a worldwide pandemic of SARS-CoV-2 disease, and the media and the government health sector gave more importance to promoting the care of this disease. Likewise, this time between evaluations and the pandemic that was experienced also affected the expected non-response rate (expected 10% vs. presented 34%). Nevertheless, the total number of participants allowed for a sufficient sample for the expected results.

Although there is robust evidence that WL is more understandable and acceptable than GDA in the Mexican population, our study stands out for being the first to be conducted at the time of the implementation of the 2020 labeling law. This may have allowed us to estimate the perception and understanding of the systems in consumers who were familiar and unfamiliar with GDA and WL. Additionally, compared to other studies assessing comprehension and acceptability of labels through an experimental design, our study assessed these variables using a two-stage panel design with the same participants and the same questionnaire structure. This allowed for a better evaluation of perception and understanding among participants, which would have required us to randomize characteristics of the study population to control for potential confounders.

Finally, our study is limited in its ability to replicate a real experience, as participants were not shown additional information such as the nutritional table and/or the list of ingredients of the products used, as well as health claims. However, fictitious products were used to reduce product familiarity by displaying the FOPL information according to the actual FOPL nutritional and nutrient quality criteria.

## 5. Conclusions

During the implementation process of the WL in Mexico, the studied population had a better perception and understanding of less healthy packaged foods when using WL, compared to GDA labels. Therefore, it appears that WLs are more effective in conveying information about the nutritional quality of foods. This evidence can be useful for public health in Mexico and beyond. If WL can help guide the purchasing decisions of Mexican consumers, it could help improve their nutritional status and prevent chronic diseases resulting from inadequate nutrition.

## Figures and Tables

**Figure 1 nutrients-14-03403-f001:**
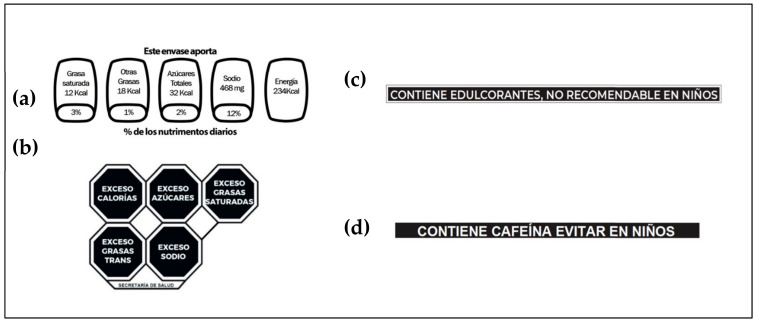
Front-of-pack label and disclaimers used: (**a**), Guideline Daily Allowance; (**b**), Warning Label octagons; (**c**), Sweetener disclaimer, and (**d**), Caffeine disclaimer.

**Figure 2 nutrients-14-03403-f002:**
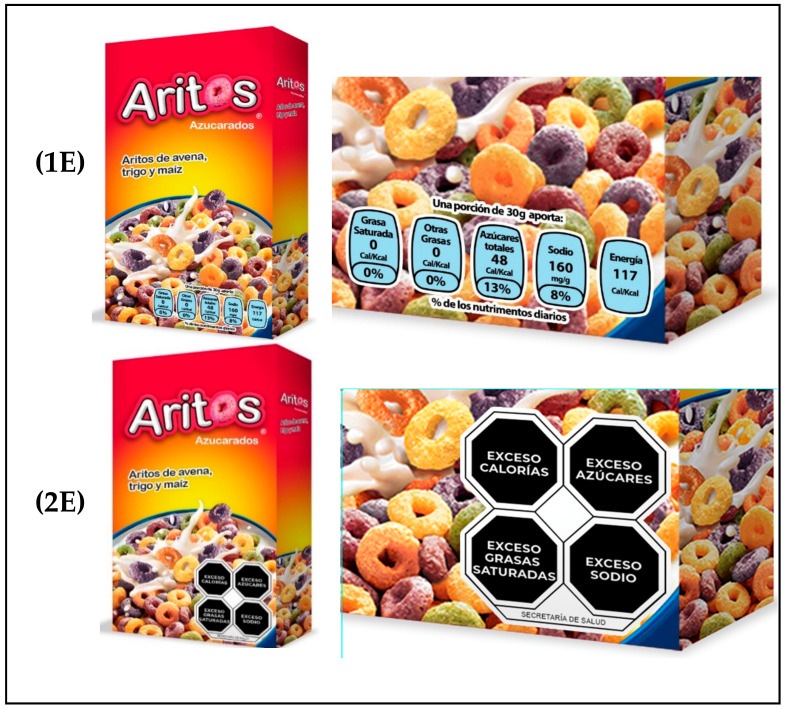
Examples of products used in survey. (**1E**), first evaluation; (**2E**), second evaluation.

**Table 1 nutrients-14-03403-t001:** Sociodemographic characteristics of the participants (*n* = 1083).

	*n*	%
**Age (years), mean** ± **SD**	36.5	15.35
**Age Tertiles (years)**		
18–25	382	32.3
26–42	329	30.4
43–75	372	34.4
**Gender (Female)**	649	60%
**Socio Economic Status**		
Low	755	69.7
Medium	328	30.3
**Academic level**		
Secondary or lower	169	15.6
High school	359	33.2
Bachelor’s degree or above	555	51.3
**BMI (kg/m^2^) ^a^**		
Normal	460	42.5
Overweight	407	37.6
Obesity	216	19.9
**Previous diagnosis**		
Hypertension	151	13.9
Diabetes	79	7.3
Overweight	457	42.2
Obesity	173	16.0
High cholesterol	153	14.1
High triglycerides	147	13.6
**Nutrition knowledge ^b^**		
Nothing knowledgeable	145	13.4
A little knowledgeable	505	46.6
Somewhat and very knowledgeable	433	40.0
**Interest in own health ^c^**		
Not or a little interested	34	3.1
Sufficiently interested	337	31.1
Very interested	712	65.7

**^a^** Body Mass Index (BMI) calculated by self-reported height and weight: <25 kg/m^2^ (normal); 25–29.9 kg/m^2^ (overweight); ≥30 kg/m^2^ (obesity); **^b^** the participants were asked, ‘How much do you think you know about nutrition?’; **^c^** the participants were asked, ‘How much are you interested in your health?’.

**Table 2 nutrients-14-03403-t002:** Differences in the perception of the GDA and WL schemes (*n* = 1083).

	*GDA ^1^*		*WL ^1^*		GDA-WL ^2^	
	*September 2020*		*October–November 2020*			
	*%*	*95%CI*	*%*	*95%CI*	*Diff pp*	*95%CI*
Would you consume a food with this label more frequently? (Unlikely- very unlikely) ^3^	24.2	21.7, 26.7	81.5	79.2, 83.8	57.3	54.0, 60.6
In your opinion, in what quantities should a product with this label be consumed? (Small or very small amounts) ^3^	41.6	38.7, 44.6	93.8	92.4, 95.5	52.2	48.9, 55.4
What would you do if you saw this label on a product that you usually buy? (Probably or very probably stop buying it) ^3^	24.4	21.8, 26.9	72.8	70.1, 75.4	48.4	44.8, 51.9
This label catches my attention (strongly agree- totally agree) ^3^	14.8	12.7, 16.9	51.6	48.7, 54.6	36.9	33.2, 40.5
If a product had this label, you could easily identify and read it (strongly agree- totally agree) ^3^	19.8	17.5, 22.2	62.6	59.7, 65.5	42.8	39.2, 46.3
I consider the information on this label to be credible and true (strongly agree- totally agree) ^3^	17.8	15.5, 20.1	53.3	50.3, 56.2	35.5	32.0, 38.9
I think this label will not help me identify healthier product (strongly disagree- totally disagree) ^3^	24.6	22.0, 27.1	47.1	44.1, 50.1	22.5	18.8, 26.3
I think this label is easy to understand (strongly agree- totally agree) ^3^	20.9	18.5, 23.3	60.4	57.5, 63.3	39.5	36.0, 43.1
This label will help me to decide quickly what products to buy (strongly agree- totally agree) ^3^	19.1	16.8, 21.4	53.1	50.2, 56.0	34.0	30.4, 37.6
This label will help me decide whether or not to buy a product (strongly agree- totally agree) ^3^	20.8	18.4, 23.2	54.3	51.4, 57.2	33.5	29.9, 37.2
This label will not change my decision about which products to buy (strongly disagree- totally disagree) ^3^	17.1	14.8, 19.3	38.8	35.9, 41.7	21.7	18.1, 25.3

GDA, Guideline Daily Amounts; WL, Warning Label; Diff pp, Difference in percentage points; SE, Standard Error. The information presented refers to the percentage of the population that responded to the category for each question and each labeling. ^1^ Adjusted percentage obtained through multinomial logistic regression models, as predictive variables included age group, study stage, and their interactions, as well as SES, nutrition knowledge, and BMI, through predictive margins; ^2^ adjusted differences in percentages points obtained through multinomial logistic regression models with GDA as a reference, as predictive variables were included age group, study stage, and their interactions, as well as SES, knowledge of nutrition, and BMI, through contrast predictive of margins; ^3^ based on scales from 1 to 7 recoded to three categories. The percentages represent the highest category of each variable.

**Table 3 nutrients-14-03403-t003:** Changes in perceptions and subjective understanding among products with GDA and WL by food category ^1^. (*n* = 1083).

Reference: GDA	Dairy	RTEC	Salty Snack	SSB	RTEF
Perception	Diff. PP ± SE	Diff. PP ± SE	Diff. PP ± SE	Diff. PP ± SE	Diff. PP ± SE
How attractive is the product for consumption? ^2^ (Attractive or very attractive)	**−4.4 ± 1.39**	**−21.1 ± 1.67**	**−14.5 ± 1.70**	**−19.0 ± 1.71**	**−4.5 ± 1.09**
How healthy is the product? ^3^ (Healthy or very healthy)	**−8.4 ± 1.11**	**−20.7 ± 1.38**	**−8.1 ± 1.18**	**−12.9 ± 1.38**	**−5.2 ± 1.00**
Would you buy this product for yourself or your family? ^2^ (Unlikely- very unlikely)	**−5.0 ± 1.11**	**−20.0 ± 1.56**	**−7.1 ± 1.43**	**−11.1 ± 1.57**	**−4.2 ± 0.95**
How often would you buy this product for yourself? ^2^ (once or twice per month or never)	**4.0 ± 1.06**	**5.8 ± 1.07**	3.2 ± 1.23	**6.0 ± 1.50**	2.2 ± 0.86
Does the label of this product provide enough information to determine if it’s healthy? ^4^ (Is not informative enough)	**−22.2 ± 1.76**	**−22.4 ± 1.71**	**−15.2 ± 1.62**	**−14.8 ± 1.67**	**−17.6 ± 1.65**
Front of pack labeling makes you feel? ^4^ (Safer to decide if the product is healthy)	**38.1 ± 1.93**	**37.4 ± 1.87**	**34.9 ± 1.90**	**34.2 ± 1.89**	**37.0 ± 1.85**
**Subjective Understanding**					
Correct identification of total number of high critical nutrients ^5^	**39.1 ± 1.71**	**74.9 ± 1.41**	**54.8 ± 1.68**	**27.7 ± 1.53**	**35.1 ± 1.74**

**Bold numbers** mean statistics differences (*p* < 0.001). GDA, Guideline Daily Amounts; WL, Warning Label; Diff PP, Difference in percentage points; SE, Standard Error; RTEC, Ready-to-eat cereals, SSB, Sugar sweet beverage; RTEF, Ready-to-eat foods. The information presented refer differences in percentage points of the population that respond to the category mentioned of each product after implementation of WL vs. GDA. ^1^ Adjusted difference in percentages points obtained through logistic regression models with GDA as a reference, as predictive variables were included age group, study stage, and their interactions, as well as SES, nutrition knowledge, and BMI, through contrast predictive of margins; ^2^ based on scales from 1 to 7 recoded to two categories. The categories presented represent the highest category of each var-iable.^3^ based on scales from 1 to 7 recoded to two categories. The categories presented represent the lowest category of each variable; ^4^ based on a scale of three categories recoded to two categories. The category presented represent the highest category of each variable; ^5^ two categories variable. The category represents the correct identification of high amounts of fat, sugars, calories/energy, and sodium.

## Data Availability

The data that support the findings of this study are available from the correspondence author upon reasonable request.

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
