# Peer review of "Perception and Understanding of Guideline Daily Amount and Warning Labeling among Mexican Adults during the Law Modification Period"

_nutrients, 2022, doi:10.3390/nu14163403_

Round 1

Reviewer 1 Report

Add reference(s) for sentence in lines 43-45.

Lines 88-89: The authors should consider re-writing this sentence; a total of at least 20 participants were recruited in the study? Or did each research assistant recruit 20 participants? The determination of the sample size is later described, but the sentence in lines 88-89 is unclear. 

What proportion of questionnaires were self-administered vs. interviewer-administered at each stage (1E and 2E)?

Line 156: Authors should briefly explain the AMAI questionnaire, and what does the “AMAI” abbreviation mean?

The authors should consider checking the English language and style. There are some typos (e.g., 95% CI and 95%CI are both used) and grammar errors. 

Author Response

Reviewer #1

Add reference(s) for sentence in lines 43-45.

R: Thank you for your observation. A reference has been added to the mentioned line.

Lines 88-89: The authors should consider re-writing this sentence; a total of at least 20 participants were recruited in the study? Or did each research assistant recruit 20 participants? The determination of the sample size is later described, but the sentence in lines 88-89 is unclear. 

R: Thank you and we agree with the recommendation. We rewrite the line that is mentioned. 

What proportion of questionnaires were self-administered vs. interviewer-administered at each stage (1E and 2E)?

R: Thanks for the comment, a mistake was made, the questionnaire was self-administered, it was made clear on line 93.

Line 156: Authors should briefly explain the AMAI questionnaire, and what does the “AMAI” abbreviation mean?

R: Thanks for the suggestion. We have added a brief explanation of the questionnaire in the section on covariates section 2.5. 

The authors should consider checking the English language and style. There are some typos (e.g., 95% CI and 95%CI are both used) and grammar errors. 

R: Thanks for the suggestion. We have sent the entire manuscript for style and grammar review.

Reviewer 2 Report

This study compares the perception and understanding of guideline daily amount and warning labels in Mexico. While there have been numerous studies on the perceptions of warning labels, this study is useful as it was conducted after the introduction of WL.

Generally this study is clearly and concisely presented but there are instances where the English needs to be revised.

Abstract: I found the abstract a bit difficult to follow and it was not until I read the paper that I understood that WL are mandatory in Mexico, and that the first evaluation was GDA and the second was WL (after implementation of WL). I realise the authors are restricted by word count but I suggest the background and methods of the abstract are rewritten to make it clearer - e.g. ... at the end of GDA and after the introduction of mandatory WL ....

The introduction is concise and relevant. 

33 - change outcomes to outcome

47 - change affect to affects

61,62 - reword this sentence. ... since Oct 2020 Mexico implemented ....

Methods

Generally the methods are clear. I suggest in the first paragraph it is clear that the first evaluation is GDA and the second is WL. 

93- Can the questionnaire be administered by an interviewer or self-administered? It is not clear.

107 - the term subjective perception is used. But isn't all perception subjective? There are questions about perceptions and an exercise to test subjective understanding. Is this what is meant here? 

186-188 I think this sentence needs an ending like 'were calculated'

Results

Generally the results are clearly presented but I wasn't clear about the Likert scale and terms used.

Table 1 - The term more or less is not clear. How does less differ from a little.

Table 2 - should be first question be 'Would you consume a food with this label more frequently?' There seems to be a couple of answer options combined for each question but this is not clear. Perhaps add the wording you used in Table 3 - The categories presented represent the x highest/lowest of each variable.

222 - The ending seems to be missing - add wording like 'were measured'

Discussion

The discussion is relevant and compares to similar studies in Mexico and other countries with discussion of the strengths and limitations of this study.

250-251 - reword - participants were more likely to understand the WL compared to nutrition facts

295 - clairvoyance is not the right word here

322-325 - This sentence is not clear to me. Can you rewrite or delete - I am not sure if it is needed.

Author Response

This study compares the perception and understanding of guideline daily amount and warning labels in Mexico. While there have been numerous studies on the perceptions of warning labels, this study is useful as it was conducted after the introduction of WL.

Generally this study is clearly and concisely presented but there are instances where the English needs to be revised.

R: Thanks for the suggestion. We have sent the entire manuscript for style and grammar review.

Abstract: I found the abstract a bit difficult to follow and it was not until I read the paper that I understood that WL are mandatory in Mexico, and that the first evaluation was GDA and the second was WL (after implementation of WL). I realise the authors are restricted by word count but I suggest the background and methods of the abstract are rewritten to make it clearer - e.g. ... at the end of GDA and after the introduction of mandatory WL ....

R: Thank you very much for the suggestion. We have rewritten much of the abstract.

The introduction is concise and relevant. 

R: Thanks

33 - change outcomes to outcome

R: Thanks for the comment, we change the word.

47 - change affect to affects

R: Thanks for the comment, we change the word.

61,62 - reword this sentence. ... since Oct 2020 Mexico implemented ....

R: Thanks for the comment, we rewrite de line.

Methods

Generally the methods are clear. I suggest in the first paragraph it is clear that the first evaluation is GDA and the second is WL. 

R: Thank you for the comment, we clarified the times and the labeling that was used.

93- Can the questionnaire be administered by an interviewer or self-administered? It is not clear.

R: Thanks for the comment, a mistake was made, the questionnaire was self-administered, it was made clear on line 93.

107 - the term subjective perception is used. But isn't all perception subjective? There are questions about perceptions and an exercise to test subjective understanding. Is this what is meant here? 

R: Thanks for the comment, yes just what the reviewer mentions is what was meant. We made that point clearer on line 107.

186-188 I think this sentence needs an ending like 'were calculated'

R: Thank you for the suggestion. We have edited the above mentioned line.

Results

Generally the results are clearly presented but I wasn't clear about the Likert scale and terms used.

R: Thank you for the comment. we have added more clarity with Likert scales on results section

Table 1 - The term more or less is not clear. How does less differ from a little.

R: Thank you for your comment, we have made it clearer what the different categories refer to.

Table 2 - should be first question be 'Would you consume a food with this label more frequently?' There seems to be a couple of answer options combined for each question but this is not clear. Perhaps add the wording you used in Table 3 - The categories presented represent the x highest/lowest of each variable.

R: Thank you for the suggestion. we have added more clarity with Likert scales at the bottom of table 3.

222 - The ending seems to be missing - add wording like 'were measured'

R: Thank you for the suggestion. We have edited the above-mentioned line.

Discussion

The discussion is relevant and compares to similar studies in Mexico and other countries with discussion of the strengths and limitations of this study.

250-251 - reword - participants were more likely to understand the WL compared to nutrition facts

R: Thank you for the suggestion. We rewrite the above-mentioned line.

295 - clairvoyance is not the right word here

R: Thanks for the comment, we have edited the word

322-325 - This sentence is not clear to me. Can you rewrite or delete - I am not sure if it is needed.

R: Thanks for the comment, we deleted the sentence.